# Can VA-ECMO Be Used as an Adequate Treatment in Massive Pulmonary Embolism?

**DOI:** 10.3390/jcm10153376

**Published:** 2021-07-30

**Authors:** Raphaël Giraud, Matthieu Laurencet, Benjamin Assouline, Amandine De Charrière, Carlo Banfi, Karim Bendjelid

**Affiliations:** 1Intensive Care Unit, Geneva University Hospitals, 1205 Geneva, Switzerland; matthieu.laurencet@gmail.com (M.L.); benjamin.assouline@hcuge.ch (B.A.); amandine.decharriere@hcuge.ch (A.D.C.); Karim.Bendjelid@hcuge.ch (K.B.); 2Geneva Hemodynamic Research Group, Faculty of Medicine, University of Geneva, 1205 Geneva, Switzerland; carbanfi@gmail.com; 3Department of Cardio-Thoracic Surgery Istituto Clinico Sant’Ambrogio, Gruppo Ospedaliero San Donato, Milan, and Chair of Cardiac Surgery, University of Milan, 20122 Milan, Italy

**Keywords:** massive acute pulmonary embolism, cardiogenic shock, VA-ECMO, thrombolysis

## Abstract

Introduction: Massive acute pulmonary embolism (MAPE) with obstructive cardiogenic shock is associated with a mortality rate of more than 50%. Venoarterial extracorporeal membrane oxygenation (VA-ECMO) has been increasingly used in refractory cardiogenic shock with very good results. In MAPE, although it is currently recommended as part of initial resuscitation, it is not yet considered a stand-alone therapy. Material and Methods: All patients with MAPE requiring the establishment of VA-ECMO and admitted to our tertiary intensive care unit were analysed over a period of 10 years. The characteristics of these patients, before, during and after ECMO were extracted and analysed. Results: A total of 36 patients were included in the present retrospective study. Overall survival was 64%. In the majority of cases, the haemodynamic and respiratory status of the patient improved significantly within the first 24 h on ECMO. The 30-day survival significantly increased when ECMO was used as stand-alone therapy (odds ratio (OR) 15.58, 95% confidence interval (CI) 2.65–91.57, *p* = 0.002). Nevertheless, when ECMO was implanted following the failure of thrombolysis, the bleeding complications were major (17 (100%) vs. 1 (5.3%) patients, *p* < 0.001) and the 30-day mortality increased significantly (OR 0.11, 95% CI 0.022–0.520, *p* = 0.006). Conclusions: The present retrospective study is certainly one of the most important in terms of the number of patients with MAPE and shock treated with VA-ECMO. This short-term mechanical circulatory support, used as a stand-alone therapy in MAPE, allows for the optimal stabilisation of patients.

## 1. Introduction

Massive acute pulmonary embolism (MAPE), defined as pulmonary embolism (PE) with sustained arterial hypotension, is associated with a high mortality rate, up to 50% [1]. The primary cause of death in MAPE is acute right ventricular failure, which induces obstructive cardiogenic shock with low systemic output. In parallel with haemodynamic and respiratory resuscitation, the treatment of MAPE consists of anticoagulation combined with reperfusion therapy.

The 2019 Guidelines of the European Society of Cardiology on the diagnosis and management of acute PE advise the use of systemic thrombolysis as a primary reperfusion therapy. In the case of contraindications to thrombolysis or the failure thereof, it is recommended to undertake surgical embolectomy (SE) or percutaneous catheter-directed treatment [2]. Venoarterial extracorporeal membrane oxygenation (VA-ECMO) may be helpful in patients with MAPE and circulatory collapse or cardiac arrest, but its use as a stand-alone technique with anticoagulation remains controversial and should be associated with a reperfusion therapy, such as SE [3].

There are several aetiologies of heart failure that are bearable with VA-ECMO. These can be broadly classified into severe and refractory forms of cardiogenic shock [4], cardiac arrest [5], refractory ventricular arrhythmias [6], acute myocarditis [7] and acute or decompensated right ventricular (RV) heart failure (as with MAPE) [8,9,10,11,12]. In MAPE, case series and cohort studies have reported that ECMO has been used as a replacement, as a complement or as a rescue technique serving as a bridge to reperfusion therapy. However, due to contraindications or major clinical instability, some patients do not lend themselves to reperfusion therapy or do not improve after this treatment [13,14,15].

VA-ECMO is one of the most reliable and fastest ways to reduce RV overload, improve RV function and haemodynamic status and restore tissue perfusion. Some authors have suggested that VA-ECMO might be used alone until preventive treatment with heparin and spontaneous endogenous thrombolysis to permit the weaning of supportive therapy [9,15].

We hereby present a retrospective cohort of patients with MAPE who were treated with VA-ECMO alone as a part of advanced life support or as a rescue treatment after a failed attempt at systemic thrombolysis and/or catheter thromboaspiration. We hypothesised that ECMO support, associated with therapeutic heparinisation, is a satisfactory treatment for MAPE.

## 2. Material and Method

### 2.1. Patients

We retrospectively analysed a cohort of patients referred to our 36-bed ICU with MAPE, which was diagnosed according to the diagnostic strategy of the European Society of Cardiology guidelines, who needed VA-ECMO support [16]. The patients were selected with the following criteria: at least 18 years old, presenting with acute PE and haemodynamic instability, under vasopressor support or undergoing thrombolysis, and between 2010 and 2019. The patients received either the standard therapy and ECMO support or ECMO support only. The data were manually extracted from the digital records of patients from the Geneva University Hospitals. The inotrope score (12) (IS) and the sequential organ failure assessment (SOFA) score were computed directly from the Patient Data Management System (Centricity Critical Care, Clinisoft, GE HealthCare, General Electric Company, Chicago, IL, USA). This study and its protocol have been approved by the Regional Research Ethics Committee (CCER 2017-00460). Informed consent was not deemed necessary given the nature of the present analyses.

### 2.2. Criteria for VA-ECMO Implantation

The criteria for VA-ECMO implantation in the setting of MAPE were decided by pulmonary embolism response teams (PERTs), including intensivists, cardiac surgeons, cardiologist, anaesthesiologists and angiologists, and are as followings: 1. refractory cardiogenic shock, defined as evidence of tissue hypoxia (e.g., elevated blood lactate or extensive skin mottling) despite adequate volemia; severe RV failure, defined as low cardiac output syndrome associated with echocardiographic signs of RV dilation, RV/LV dimension ration > 1.5 and septal flattering; low cardiac index (≤2.1 L/min/m^2^) under inotropes; sustained hypotension despite high-dose vasoactive drug infusion (norepinephrine ≥ 0.3 µg/kg/min, epinephrine ≥ 0.3 µg/kg/min or dobutamine ≥ 10 µg/kg/min); or 2. refractory cardiac arrest (no ROSC after 30 min of advanced resuscitation) with no-flow time below 5 min, low-flow time below 100 min, or effective external chest compressions with an end-tidal CO_2_ (EtCO_2_) > 1.5 kPa. ECMO exclusion criteria were age > 80 years old, malignancies with poor prognosis within 2 years or irreversible and disabling neurological pathologies and decisions to limit therapeutic interventions.

### 2.3. VA-ECMO Cannulation and Management

VA-ECMO was percutaneously or surgically implanted by trained intensivists, interventional cardiologists or cardiovascular surgeons with femoro-femoral cannulae from 25 to 28 French for the venous cannula and from 17 to 21 French for the arterial cannula, as previously described [17]. An additional 6- to 7-French catheter was systematically inserted into the superficial femoral artery to perfuse the lower leg and prevent tissue ischaemia. In the case of highly unstable and non-transportable patients, our mobile ECMO team travelled to primary care hospitals with a portable ECMO system, implanted the device at bedside in the ICU and transported the patient to our ICU [18]. The ECMO pump speed was adjusted to obtain blood flow between 3.5 and 5 L/min and therapeutic anticoagulation was perfused with intravenous UHF to maintain an anti-factor Xa (anti-Xa) level between 0.3 and 0.6 UI/L.

### 2.4. Data Collection

After ICU admission, the following variables were collected: demographics (age, gender, and body mass index); simplified acute physiology score (SAPS) II; haemodynamic status before intervention (cardiac arrest, no-flow and low-flow times, pre-intervention systolic blood pressure, mean blood pressure, heart rate and IS and shock onset-to-intervention interval); blood gas analyses (pH, blood lactate and bicarbonate levels and PaO_2_/FiO_2_ ratio); general blood analyses (creatinine, aspartate and alanine aminotransferases, prothrombin time and factor V activity); specific markers of cardiac function (NT-proBNP or pro-BNP and troponin I or troponin T); radiographic exams (ultrasound and computed tomography (CT)); sonographic findings, if any (RV dilation, pulmonary thrombus visible, left ventricular ejection fraction and tricuspid annular plane systolic excursion); chest CT findings, if any (proximal PE, pulmonary infarction and right ventricle-to-left ventricle diameter ratio); and previous reperfusion therapeutic interventions, if any (systemic fibrinolytic therapy, surgical thrombectomy, catheter-directed thromboaspiration, pulmonary angioplasty and surgical endarterectomy). After 24 h, the IS, pH and blood lactate levels were extracted. At discharge from the ICU, the quantity of blood products transfused during the ICU stay (packed red blood cells (PRBC), fresh frozen plasma (FFP) and thrombocytes) were also extracted. Mechanical ventilation duration, ICU and hospital lengths of stay and mortality at 30 days were also obtained. Average ECMO blood flow during the first 24 h, as well as ECMO duration, ECMO-related complications and post-ECMO information, were collected. Bleeding complications were reported using the global utilisation of streptokinase and TPA for occluded arteries (GUSTO) classification [19]. Severe life-threatening haemorrhage was defined as intracerebral bleeding or bleeding resulting in substantial haemodynamic compromise requiring treatment (GUSTO 1). GUSTO 2 defined moderate bleeding as the need for transfusion, whereas GUSTO 3 referred to other bleeding, not requiring transfusion or causing haemodynamic compromise.

Finally, for survivors, at one year, the presence or absence of chronic dyspnoea or chronic thromboembolic pulmonary hypertension (CTEPH) were determined and collected.

### 2.5. Statistical Analyses

Results are presented as median and interquartile ranges (IQR) or numbers with percentages. Continuous variables were compared using the nonparametric Mann–Whitney U test or Wilcoxon signed-rank test as appropriate. Categorical variables were analysed with Fischer’s exact test or the Chi-squared test as appropriate. Logistic regression analysis was performed to determine independent predictors associated with 30-day survival. Variables with significant associations using univariate analysis (*p* < 0.05) were entered into the multivariate analysis. Odds ratios (OR) and 95% confidence intervals (CI) were also calculated. Kaplan–Meier curves for the 30-day survival of patients with MAPE implanted with only an ECMO standalone or treated with fibrinolysis and/or catheter thromboaspiration before ECMO implantation were compared with a log-rank test and a Cox proportional hazards model. Statistical analyses were performed with STATA (StataCorp. 2019. Stata Statistical Software: Release 16. College Station, TX, USA: StataCorp LLC.), and a two-sided *p* < 0.05 was considered significant.

## 3. Results

During the 10-year period of the study, 36 patients (27 males (75%); median age 57 (IQR 23)) presenting with MAPE received VA-ECMO. Table 1 displays the description of the complete cohort. Thirty (83.3%) patients had predisposing factors for venous thromboembolism. Twenty-two (61.7%) patients presented with cardiac arrest—all with a no-flow time of zero minutes and a low-flow time of 28 ± 30 min. Thirteen (36.1%) patients received ECMO as part of advanced cardiopulmonary resuscitation (eCPR). Accordingly, before ECMO institution, both median systolic arterial (74 mmHg (IQR 54)) and mean arterial (59 mmHg (IQR 35)) blood pressures were low. Mean values for the acid-base balance status reflected this situation with a low pH (7.08 (IQR 0.38)), high blood lactate (8.3 mmol/L (IQR 11.1)) and low bicarbonate (15.2 mmol/L (IQR 8)).

All patients had cardiac echocardiography, whereas chest CT was only performed in nine (25%) patients, as the majority were too unstable for a radiographic exam. In the cohort, RV dilation, which is an indicator of elevated risk of short-term mortality, was found in all patients. For patients who were too unstable, thoracic CT angiography was performed a posteriori with assistance and confirming the diagnosis of PE in all cases.

The installation of the ECMO was carried out relatively quickly, since the onset of shock to the ECMO interval was 1 h (IQR 1.5). At ECMO cannulation, the median SAPS II score and IS were high, as all patients required haemodynamic support with vasoactive drugs.

Reperfusion therapy with systemic thrombolysis was undertaken in 16 (44.4%) patients, whereas 21 (58.3%) patients had contraindications to thrombolysis. Five (15.6%) patients benefited from catheter-directed thromboaspiration and none underwent surgical thrombectomy, as this latest therapy is not implemented in our hospital. All patients received peripheral femoro-femoral VA-ECMO, including 25 (69.4%) patients percutaneously.

As described in Table 2, 24 h on ECMO rapidly corrected pH, blood lactate level and IS (Figure 1).

Table 3 describes the ICU events and outcomes among ECMO-treated patients with MAPE according to 30-day survival status. When comparing non-survivors to survivors, significantly higher SAPS II scores, pre-ECMO IS and IS after 24 h of ECMO were found in non-survivors. ECMO duration and both ICU and hospital lengths of stay were shorter among non-survivors. For in-ICU ECMO-related complications, haemorrhage levels and PRBC units transfused were also significantly higher in non-survivors.

The variables independently associated with 30-day survival status among ECMO-treated patients are presented in Table 4. A univariate analysis showed that ECMO as a stand-alone therapy was highly associated with 30-day survival (OR 15.58, 95% CI 2.65–91.57, *p* = 0.002). However, thrombolysis failure, ECMO during cardiopulmonary resuscitation vs. not, ECMO and fibrinolytic treatment or catheter-directed thromboaspiration vs. ECMO alone and the presence of cardiac arrest before ECMO initiation vs. not were found to significantly increase the risk of death.

ECMO-related complications and 30-day survival status, compared between patients treated with ECMO only vs. patients who received ECMO + fibrinolytic treatment, are presented in Table 5. Haemorrhaging was significantly higher in patients receiving ECMO + fibrinolytic or catheter-directed thromboaspiration vs. ECMO alone. Thirty-day mortality was also significantly higher in patients receiving ECMO + fibrinolytic or catheter-directed thromboaspiration vs. ECMO alone.

Kaplan–Meier curves of patients with MAPE implanted with only an ECMO standalone or treated with fibrinolysis and/or catheter thromboaspiration before ECMO implantation showed a significant difference in 30-day survival (Cox proportional hazards model: *p* = 0.004, HR 8.563 (95% CI 2.775–25.76) (Figure 2).

Finally, all patients with anoxic encephalopathy in relation to cardiac arrest died. At the one-year follow-up, only one patient (2.8%) was diagnosed with chronic dyspnoea and one patient (2.8%) had CTEPH.

## 4. Discussion

To our knowledge, this is one of the largest follow-up studies on life-threatening MAPE treated with VA-ECMO. The main finding of the present study is that ECMO as a stand-alone therapy for MAPE is associated with a decrease in 30-day mortality in comparison with ECMO + fibrinolytic or catheter-directed thromboaspiration. Moreover, ECMO during cardiopulmonary resuscitation and pre-ECMO cardiac arrest with ROSC are both associated with an increase in 30-day mortality. Finally, the association of ECMO + fibrinolytic or catheter-directed thromboaspiration is correlated with more haemorrhagic complications compared with ECMO as a stand-alone therapy for MAPE.

In MAPE, mechanical obstruction due to thrombus present in the pulmonary arteries, associated with hypoxic pulmonary vasoconstriction and the release of vasoconstrictor mediators, leads to a sharp increase in the RV afterload [13]. It results in dilation with subsequent RV failure and a deviation of the intraventricular septum to the disadvantage of the left ventricle with a decrease in its preload and a drop in cardiac output, inducing cardiogenic shock with hypotension and hypoperfusion of tissues, as well as cardiac arrest in some cases [20,21]. In parallel with the haemodynamic impairment, MAPE induces respiratory failure. A low level of cardiac output causes desaturation of mixed venous blood. Areas of reduced flow in the obstructed pulmonary arteries, associated with areas of overflow in the capillary bed served by unobstructed pulmonary vessels, cause a ventilation/perfusion mismatch, which contributes to hypoxaemia [22] and an increase in dead space. ECMO is now recommended by the latest Guidelines of the European Society of Cardiology (ESC) as a temporary mechanical circulatory support in patients with MAPE and circulatory collapse or cardiac arrest [2], offering full cardiopulmonary support [23], as shown by our results. Indeed, pH, blood lactate level and IS significantly improved 24 h after ECMO implantation.

Although no RCTs testing the efficacy and safety of these devices in the context of MAPE have been conducted to date, the survival of critically ill patients has been described in a number of case series [3,9,12,24,25,26]. The main explanation given is that ECMO is associated with a high incidence of complications, particularly an increased risk of bleeding related to the need for vascular access and mostly in patients in whom systemic thrombolysis failed. The latter is confirmed by our study. However, our results did not show more haemorrhagic complications (22%) compared with a study evaluating 304 patients from the International Cooperative Pulmonary Embolism Registry who received fibrinolysis, showing that 66 patients (21.7%) had major bleeding [27], or data from a tertiary centre in Paris, showing that of 132 patients receiving thrombolysis, 33 (25%) experienced major bleeding [28].

For many years, the recommended reperfusion therapy for PE has been systemic thrombolysis, with IB-level evidence [2,13]. In the latest meta-analysis, on which most of these recommendations are based, only four studies included patients with MAPE [29]. Among these, three were published before 1980. The most recent study included in the meta-analysis dates from 1995 and included only eight patients [30]. Although the results showed that the four patients who received streptokinase survived, whereas the four patients who received heparin died, no conclusion or recommendations can be made, neither on the basis of this study nor on the results of the meta-analysis concerning patients with shock and MAPE. In addition, most studies support the fact that systemic thrombolysis increases the risk of bleeding, particularly in patients with cardiogenic shock and cardiac arrest [29], except a recent systematic review about 301 patients, showing that 51 patients who received systemic thrombolysis prior to cannulation had similar survival compared with patients who did not (67% vs. 61%, respectively; *p* = 0.48) [31].

Thrombolysis, in addition to having many contraindications, is not always effective. In our study, 21 (58.3%) patients had an absolute contraindication to systemic thrombolysis, and in the other 15 (41.6%) patients, ECMO was implemented in a context of failed thrombolysis. In the 18 (50%) patients in our cohort who presented with bleeding, all had received systemic thrombolysis. Therefore, the failure of thrombolysis appears to be a factor that is directly correlated with an increase in 30-day mortality. In the study by Maggio et al., which included 21 patients (19 of whom had been implanted with VA-ECMO for MAPE), the 11 (52%) patients who underwent treatment before cannulation, such as thrombolytic agents, catheter-based intervention or surgical pulmonary embolectomy, failed. Of these 11 patients, there was only a 45% survival rate. However, in the 10 patients in whom ECMO was initiated as the primary, initial intervention, there was an 80% survival rate [8].

Given patient heterogeneity, the multitude of available treatment modalities and the lack of consensus guidelines, treatment strategies for these complex patients with MAPE are often debated. Multidisciplinary pulmonary embolism response teams (PERTs) are dedicated to address this lack of consensus [32]. An analysis among 769 consecutive inpatients with PE was conducted to compare the outcomes of all patients with PE before and after PERT availability. The results showed that PERT-era patients had lower rates of major or clinically relevant non-major bleeding (17.0% vs. 8.3%, *p* = 0.002) and shorter times to therapeutic anticoagulation (16.3 h vs. 12.6 h, *p* = 0.009). There was also a significant decrease in 30-day/inpatient mortality (8.5% vs. 4.7%, *p* = 0.03), particularly in patients with MAPE (mortality 10.0% vs. 5.3%, *p* = 0.02) [33]. As in our centre, the availability of multidisciplinary PERT certainly allows improved outcomes, including 30-day mortality.

Despite very strict implantation criteria (all the patients in this study had, in particular, a no-flow time of 0 min), the occurrence of cardiac arrest in the context of a massive PE has a poor prognosis. In a consecutive cohort of thirteen patients with confirmed MAPE for whom PERT was activated and selected patients treated with ECMO confirmed that patients with MAPE who suffer a cardiac arrest have high morbidity and mortality [34]. This was confirmed in a systematic review, in which multivariate analysis showed a six-fold increase in the risk of death if cannulation occurred during cardiopulmonary resuscitation (adjusted odds ratio, 5.67; *p* = 0.03) [31]. The current recommendations emphasise that starting a high-quality cardiac massage as quickly as possible influences the effectiveness of all other interventions [35]. Therefore, it is essential that CPR be started immediately after the collapse in order to keep the no-flow period to a minimum. It is therefore essential that the CPR is of good quality during the low-flow period [14,23]. To ensure this, it is recommended to monitor expired CO_2_ (EtCO_2_), which is a validated predictor of survival in CA [13].

In our study, no patient underwent surgical embolectomy; however, our survival results are quite good. As much as it seems reasonable to operate on stable patients, without cardiogenic shock and with a contraindication to thrombolysis, it seems very risky to perform this operation in patients in a state of profound shock who did not respond favourably to thrombolysis with collapsed coagulation factors. Finally, the latest published series emphasises the place of ECMO before this type of intervention, which makes it possible to stabilise patients, to perform surgery in better conditions and to ensure optimal postoperative haemodynamic support [36]. It is therefore questionable whether to perform SE while the patient is stabilised on VA-ECMO with good organ perfusion, rather than waiting for the thrombus and RV failure to resolve thanks to the thrombolytic properties of heparin and to natural endogenous fibrinolysis, allowing the weaning of circulatory support [9,14].

The strength of this study is the number of patients analysed with two treatment options and their descriptions. However, our study also presents some limitations. First, it is a single-centre, retrospective study, focusing only on patients receiving an ECMO in the context of MAPE without comparing them to patients who have been optimally treated with reperfusion therapy. Second, the number of patients included remains limited, thus reducing the power of the statistical analysis. Third, there was no prospective follow-up after hospital discharge based on long-term cardiac echocardiography and imaging to detect the potential development of CTEPH in the absence of thrombus extraction (by thrombolysis or SE). Further studies focusing on this point are needed to support the long-term safety of an ECMO strategy without additional mechanical clot removal therapies.

## 5. Conclusions

The present retrospective monocentric study is certainly one of the most standardised in terms of the number of patients with MAPE who presented with refractory cardiogenic shock and who were put on VA-ECMO. Overall survival was almost 64%. Our results show that this short-term mechanical circulatory support allows for the optimal stabilisation of severely ill patients. In addition, when used as a stand-alone therapy, it allows for a significant improvement in 30-day survival. Nevertheless, when it is implanted following a failure of thrombolysis, the bleeding complications are major and the 30-day mortality increases significantly. The main causes of death are the ECMO-thrombolysis association, the presence of cardiac arrest and the implementation of ECMO during cardiac arrest. Prospective studies are definitely required in order to clarify the role of ECMO in MAPE.

## Figures and Tables

**Figure 1 jcm-10-03376-f001:**
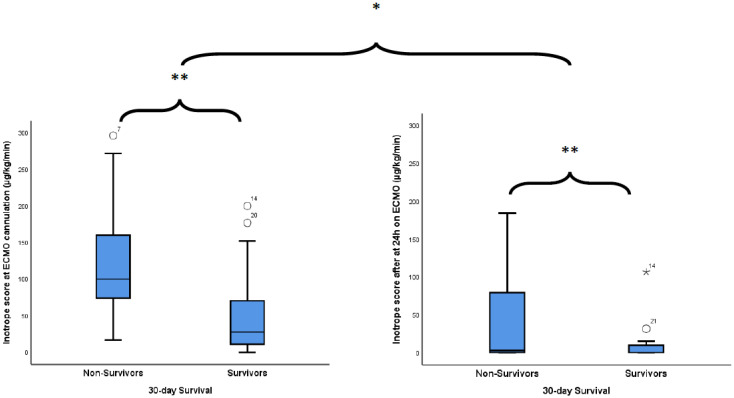
Inotrope score change between pre and post-VA-ECMO cannulation according to patients’ 30-day status. * *p* < 0.001 between pre-ECMO and after 24 h on ECMO, ** *p* = 0.004 between survivors and non-survivors.

**Figure 2 jcm-10-03376-f002:**
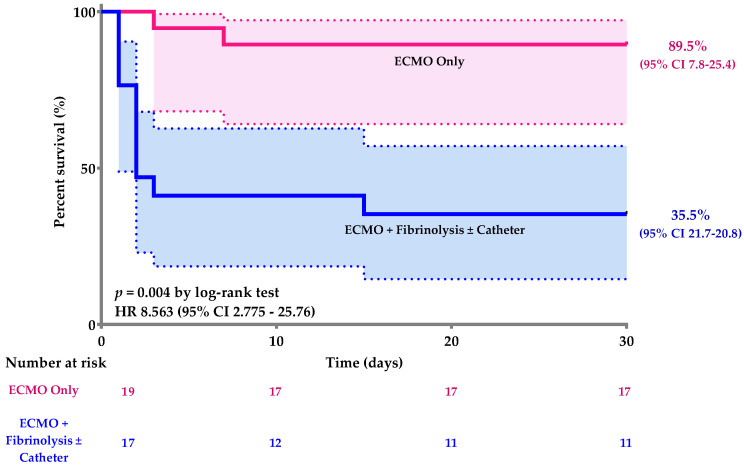
Kaplan–Meier curves for the 30-day survival of patients with MAPE implanted with only an ECMO or treated with fibrinolysis and/or catheter thromboaspiration before ECMO implantation. HR: Hazard Ratio.

**Table 1 jcm-10-03376-t001:** Description of the study population.

Variables	*n* = 36
Age, years, median ± interquartile range (IQR)	57 (23)
Male, *n* (%)	27 (75)
BMI, kg/m^2^, median (IQR)	27.8 (6.3)
Comorbidities, *n* (%)	
None	5 (14)
Postoperative (orthopaedic/visceral/other surgery/polytraumatism)	18 (50)
Medical (HBP, CKI, obesity)	5 (13.9)
Recent stroke (ischemic or haemorrhagic)	3 (8.3)
Other	5 (13.9)
Pre-ECMO	
Cardiac arrest, *n* (%)	22 (61.1)
Systemic fibrinolytic therapy, *n* (%)	16 (44.4)
Catheter-directed thromboaspiration, *n* (%)	5 (13.9)
No-flow time, min, median (IQR)	0 (0)
Low-flow time, min, median (IQR)	17.5 (52)
Systolic blood pressure, mmHg, median (IQR)	76 (54)
Mean blood pressure, mmHg, median (IQR)	59 (35)
Heart rate, bpm, median (IQR)	108 (47)
pH, median (IQR)	7.08 (0.38)
Blood lactate, mmol/L, median (IQR)	8.3 (11.1)
Bicarbonate, mmol/L, median (IQR)	15.2 (8)
PaO_2_/FiO_2_ ratio, mmHg, median (IQR)	69 (28)
Creatinine, μmol/L, median (IQR)	127 (62)
ASAT, U/L, median (IQR)	219 (416)
ALAT, U/L, median (IQR)	171 (377)
Quick, %, median (IQR)	52 (47)
Cardiac echocardiography, *n* (%)	36 (100)
RV dilation, *n* (%)	36 (100)
RV/LV dimensions ratio, cm, median (IQR)	2.1 (0.6)
TAPSE, mm, median (IQR)	8 (5.2)
Chest CT-Scan, *n* (%)	9 (25)
Surgical thrombectomy, *n* (%)	0 (0)
ECMO during cardiopulmonary resuscitation *n* (%)	13 (36.1)
Inotrope score at ECMO cannulation, µg/kg/min, median (IQR)	49 (98)
Shock onset-to-ECMO interval, hours, median (IQR)	1 (1.5)
Femoral–femoral VA-ECMO, *n* (%)	36 (100)
Percutaneous VA-ECMO, *n* (%)	25 (69.4)
Thrombolysis contraindication, *n* (%)	21 (58.3)
SAPS II at ICU admission, median (IQR)	68 (38)
One year follow-up	
Chronic dyspnoea *n* (%)	1 (2.8)
Chronic thromboembolic pulmonary hypertension *n* (%)	0 (0)
Pulmonary angioplasty, *n* (%)	1 (2.8)
Surgical endarterectomy, *n* (%)	1 (2.8)

HBP: high blood pressure, CKI: chronic kidney injury, ASAT, aspartate aminotransferase, ALAT: alanine aminotransferase, RV: right ventricle, LV: left ventricle, TAPSE: tricuspid annular plane systolic excursion.

**Table 2 jcm-10-03376-t002:** pH, Blood lactate and inotrope score evolution before and 24 h after ECMO.

	Timing	Pre-ECMO	After 24 h on ECMO	*p*-Value
Variables				
pH, median (IQR)		7.08 (0.38)	7.43 (0.1)	<0.001 ^1^
Blood lactate, mmol/L, median (IQR)		8.3 (11.1)	1.1 (0.9)	<0.001 ^1^
Inotrope score, μg/kg/min, median (IQR)		49 (98)	0 (10)	<0.001 ^1^

^1^ Wilcoxon Signed Ranks Test.

**Table 3 jcm-10-03376-t003:** ICU events and outcomes among ECMO-treated patients for massive PE according to 30-day survival status.

Variables	All Patients (*n* = 36)	Non-Survivors (*n* = 13)	Survivors (*n* = 23)	*p*-Value
**Pre-ECMO**				
**Inotrope score, μg/kg/min, median (IQR)**	**49 (98)**	**100 (86)**	**28 (60)**	**0.004 ^1^**
**Thrombolysis, *n* (%)**	**13 (36.1)**	**10 (76.9)**	**6 (26.1)**	**0.005**
pH, **median (IQR)**	7.08 (0.38)	6.99 (0.31)	7.12 (0.42)	0.055 ^1^
**Blood lactate, mmol/L, median (IQR)**	**8.3 (11.1)**	**13.8 (8.5)**	**4.2 (8.6)**	**0.008 ^1^**
**Bicarbonate, mmol/L, median (IQR)**	15.2 (8)	**11 (7.1)**	**17.1 (7.2)**	**0.018 ^1^**
PaO_2_/FiO_2_, mmHg, **median (IQR)**	69 (28)	61 (19)	73 (37)	0.415 ^1^
Creatinine, µmol/L, **median (IQR)**	127 (62)	137 (77)	124 (47)	0.214 ^1^
**ASAT, U/L, median (IQR)**	**219 (416)**	**342 (538)**	**125 (305)**	**0.015 ^1^**
**ALAT, U/L, median (IQR)**	**171 (377)**	**309 (398)**	**102 (243)**	**0.026 ^1^**
**Quick, %, median (IQR)**	**52 (47)**	**43 (27)**	**57 (52)**	**0.034 ^1^**
TAPSE, mm, **median (IQR)**	8 (5.2)	8 (5)	8.1 (4.5)	0.922 ^1^
RV/LV dimension ratio, **median (IQR)**	2.1 (0.6)	2.32 (0.84)	2.1 (0.36)	0.626 ^1^
**No-flow time, min,** **median (IQR)**	**0 (0)**	**0 (0)**	**0 (0)**	**1.000**
**Low-flow time, min,** **median (IQR)**	**17.5 (52)**	**50 (30)**	**0 (25)**	**0.002 ^1^**
**ECMO during cardiopulmonary resuscitation (vs. not), *n* (%)**	**13 (36.1)**	**8 (61.5)**	**5 (21.7)**	**0.030 ^3^**
**SAPS II at ICU admission, median (IQR)**	**68 (38)**	**75 (19)**	**55 (42)**	**0.020 ^1^**
**Inotrope score after 24 h of ECMO, μg/kg/min, median (IQR)**	**49 (98)**	**3 (44)**	**0 (10)**	**0.004 ^1^**
In-ICU complications, *n* (%)				
**Haemorrhage**	**18 (50)**	**12 (92.3)**	**6 (26.1)**	**<0.001 ^3^**
Stroke	4 (11.1)	1 (7.7)	3 (13)	0.541 ^2^
Infection	3 (8.33)	1 (7.7)	2 (8.7)	0.709 ^2^
**Packed red-cell units transfused, *n*, median (IQR)**	**4.5 (8.5)**	**8 (7)**	**0 (5)**	**0.003 ^1^**
**Fresh-frozen plasma units transfused, *n*, mean ± SD**	**1 (4)**	**5 (11)**	**0 (2)**	**0.012 ^1^**
Platelets transfused, *n*, mean ± SD	1 (5)	0 (1)	0 (0)	0.253 ^1^
**ECMO duration, days, mean ± SD**	**3.2 (3.2)**	**1 (1.3)**	**4 (2.4)**	**<0.001 ^1^**
MV duration, days, median (IQR)	2.4 (10.5)	1 (1.3)	8 (14)	0.149 ^1^
**ICU LOS, days, median (IQR)**	**8.7 (16)**	**1.2 (1.9)**	**15.8 (13.4)**	**<0.001 ^1^**
**Hospital LOS, days, median (IQR)**	**15 (36)**	**1.3 (2)**	**30 (44.5)**	**<0.001 ^1^**

^1^ For continuous variables, nonparametric Mann–Whitney test; ^2^ for categorical variables, Fischer exact test; ^3^ Chi-squared test. Bold has been used to highlight the variables statiscally significant.

**Table 4 jcm-10-03376-t004:** Variables independently associated with 30-day survival status among ECMO-treated patients for massive PE.

Variables	OR	95%CI	*p*-Value ^1^
ECMO only (vs ECMO + thrombolysis or catheter directed thromboaspiration)	15.583	2.652–91.572	0.002
Thrombolysis failure	0.106	0.022–0.520	0.006
ECMO during cardiopulmonary resuscitation (vs. not)	0.174	0.039–0.773	0.022
Pre-ECMO cardia arrest (vs. not)	0.064	0.007–0.579	0.014

^1^ Logistic regression model.

**Table 5 jcm-10-03376-t005:** ICU events (complications) and 30-day survival status between ECMO-treated patients for massive PE versus ECMO + fibrinolytic therapy.

	Exposure	ECMO Only (*n* = 19)	ECMO + Fibrinolytic or Catheter-Directed Thromboaspiration (*n* = 17)	*p*-Value
Outcomes				
In-ICU complications, *n* (%)				
**Haemorrhage**		**1 (5.3)**	**17 (100)**	**<0.001 ^1^**
Stroke		2 (10.5)	2 (11.8)	1.000 ^2^
Infection		1 (5.3)	2 (11.8)	0.593 ^2^
**Anoxic encephalopathy**		**4 (28.6)**	**10 (71.4)**	**0.039 ^2^**
**30-day death status, *n* (%)**		**2 (10.5)**	**11 (64.7)**	**0.001 ^1^**

^1^ Chi-2 test; ^2^ Fischer exact test.

## Data Availability

The datasets used and analysed during the current study are available from the corresponding author on reasonable request.

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
