# Peer review of "Can VA-ECMO Be Used as an Adequate Treatment in Massive Pulmonary Embolism?"

_jcm, 2021, doi:10.3390/jcm10153376_

Round 1

Reviewer 1 Report

Many thanks for the opportunity to review this interesting manuscript. The authors are congratulated on their experience and good outcomes. The subject is important as there is only a small amount of data on the role of VA ECMO in PE and the question of timing is challenging. An attempt to understand the impact of ECMO versus reperfusion therapy is welcome.

However, I find a number of limitations in this study that are not fully addressed and which therefore require revision:

Selection bias:

All of the patients in the study received VA ECMO. We do not know how many patients with MAPE received reperfusion therapy and improved without requiring ECMO. Only those patients who failed reperfusion therapy are represented in the comparator and so it is likely that these are more refractory cases. Therefore one cannot conclude that early ECMO is better than an initial attempt at reperfusion which is the practical implication of the study. Ideally the authors should provide data on all patients with MAPE including those who did not receive ECMO, or at least acknowledge this substantial limitation.

Confounding:

It is not known if the baseline characteristics of the ECMO vs ECMO after reperfusion groups are similar or not. It is likely that they are not.  This is not transparent and it does not appear that the multi variate logistic regression attempts to control for this. May I recommend that Table one is adapted to present data according to the primary exposure (ECMO alone vs ECMO after reperfusion) and that a full univariate analysis is presented to explore all of the factors associated with the primary outcome. The rationale for the inclusion of these factors should be given and should include pre-ECMO variables from both table 1 and table 3.

Please can the authors clarify if any of the 'ECMO alone' patients subsequently received systemic or catheter directed thrombolysis after ECMO implantation as this would be reasonable practice in some centres and would significantly influence the interpretation of the results. 

Other methodological issues:

 The multi variate logistic regression requires review in my opinion. Of the five variables presented there is substantial overlap or association between them (pre-ECMO cardiac arrest and ECMO during CPR; thrombolysis failure is essentially synonymous with ECMO after thrombolysis in this study) and two of the variables appear to be the direct corollaries of each other (ECMO vs ECMO after thrombolysis AND ECMO after thrombolysis vs ECMO). This does not appear to be robust. Additional factors that I would expect to be considered would be demographic variables such as age, markers of sickness severity (IS, lactate etc) and time to ECMO implantation. Even if these turn out not to be explanatory variables their inclusion makes the analysis and conclusion much more robust.

The authors report mean values but then perform non-parametric tests. The sample size is very small and it seems unlikely that all of the variables reported are normally distributed. I recommend reporting median values. The method only describes tests for unpaired data but then paired tests (Wilcoxon) are also reported - the method should be updated

Definitions should be made more clear please: e.g. i) what do the authors consider to be 'absolute contraindications' to thrombolysis - this should be explicit to allow readers to consider whether they agree as some contraindications may in fact be considered to be relative. ii) criteria for VA ECMO. 1. CS: are these AND, or OR criteria iii) how did you define 'RV failure' or 'high dose vasoactives' iv) haemorrhage - the finding of more haemorrhage in patients who receive thrombolysis  is important data but we need to know that this is robustly and objectively defined please.

In the tables please provide the number of patients / data points that the values are based upon / including reporting of missing data. For example, the text refers to missing data from troponin but gives no further detail. In a retrospective study of emergency interventions I would expect there to be some missing data and how this si handled is important. 

Minor comments:

The manuscript would benefit from a thorough proof read. Examples include:

Discrepancy between definition of groups in the introduction (Ln 63 - ECMO after thrombolysis) vs in the method (Ln 135-6 - ECMO after fibrinolysis or catheter aspiration)

Ln 56 - formatting of references

Ln n147 - instauration = institution

Tables: please include abbreviations in full in a legend

There is a discrepancy between the heart rate value presented in the text (Ln 148) and in Table 1

TAPSE appears to be presented as cm when the values suggest should be mm

There is a lot of duplication of results in the text that already exist in tables and this could be revised

A few abbreviations are introduced without prior explanation (e.g 'CO' Ln 236, 'ESC' Ln 240)

The discussion would benefit from some revision to make it more focused and to avoid some apparent confusions. e.g. i) The discussion about surgical embolectomy does not follow from this study as there is no data at all about this treatment (Ln291-302) ii) Ln 283-4 - the conclusion about mortality being associated with failed thrombolysis does not naturally follow from the prior discussion iii) Ln. 320 - the factors listed are not causes of death but are factors that associate with death

Respectfully, I think the discussion of weaknesses ought be expanded given the material issues identified above, unless they can be resolved to show more robustly that the conclusion of the study is robust.

Author Response

Dear Professor Bendjelid

Revised version R1- jcm-1314069: Can VA-ECMO Be Used as an Adequate Treatment in Massive Pulmonary Embolism?

We thank you for the careful review of our manuscript and for the constructive criticisms. The suggestions were very helpful and we appreciated them. We have revised the manuscript as suggested.

You will find in attached files our point-by-point responses to the comments of the reviewers. We also include a revised version of our manuscript, containing a document with changes highlighted in red.

All authors have contributed significantly to the present work. They have read and approved submission of the manuscript,

We thank you by advance for reconsidering our manuscript and hope that you will now find it suitable for publication in Journal of Clinical Medicine. Additionally, we thank you for your generous consideration.

Looking forward to hearing from you.

Raphael Giraud, MD, PhD

Reviewer 1

Many thanks for the opportunity to review this interesting manuscript. The authors are congratulated on their experience and good outcomes. The subject is important as there is only a small amount of data on the role of VA ECMO in PE and the question of timing is challenging. An attempt to understand the impact of ECMO versus reperfusion therapy is welcome.

However, I find a number of limitations in this study that are not fully addressed and which therefore require revision:

Selection bias :

All of the patients in the study received VA ECMO. We do not know how many patients with MAPE received reperfusion therapy and improved without requiring ECMO. Only those patients who failed reperfusion therapy are represented in the comparator and so it is likely that these are more refractory cases. Therefore one cannot conclude that early ECMO is better than an initial attempt at reperfusion which is the practical implication of the study. Ideally the authors should provide data on all patients with MAPE including those who did not receive ECMO, or at least acknowledge this substantial limitation.

The authors thank the reviewer for her/his comment. The present study focuses on patients receiving ECMO in the context of MAPE. We acknowledge this limitation and this has been added in the limitation paragraph as follow: “…our study also presents some limitations. First, it is a single-centre, retrospective study, focusing only on patients receiving an ECMO in the context of MAPE without comparing them to patients who have been optimally treated with reperfusion therapy.”

Confounding :

It is not known if the baseline characteristics of the ECMO vs ECMO after reperfusion groups are similar or not. It is likely that they are not. 

The authors thank the reviewer for her/his valuable comment. The present study is retrospective and concern a cohort of patients who received a VA-ECMO in the context of MAPE. It is not designed to compare to types of therapies (ie ECMO only vs ECMO associated with other therapies). Thus, it is very unlikely that these two groups are similar. However, this study is able to display a comprehensive picture of the differences in outcome between patients who received ECMO only and patients who received ECMO + another therapy.

This is not transparent and it does not appear that the multi variate logistic regression attempts to control for this.

The authors thank the reviewer for her/his comment. Concerning variables independently associated with 30-day survival status among ECMO-treated patients for massive PE, when we assessed the five variables (ECMO only, Thrombolysis failure, ECMO during cardiopulmonary resuscitation, ECMO + fibrinolytic drugs and pre-ECMO cardiac arrest), all were significantly associated to 30-day mortality. When we put all the five variables, of the present small sample size, in the same regression model, none of them stayed significantly associated with 30-day mortality. For this reason, no multivariate model is able to be presented in this manuscript.

May I recommend that Table one is adapted to present data according to the primary exposure (ECMO alone vs ECMO after reperfusion) and that a full univariate analysis is presented to explore all of the factors associated with the primary outcome.

The authors thank the reviewer for her/his comment. Only the variables independently associated with 30-day survival status among ECMO-treated patients for massive PE are presented in table 4. The other variables are not independently associated with 30-day survival and thus not reported in the table.

The rationale for the inclusion of these factors should be given and should include pre-ECMO variables from both table 1 and table 3.

The authors thank the reviewer for her/his comment. Only the variables independently associated with 30-day survival status among ECMO-treated patients for massive PE are presented.

Please can the authors clarify if any of the ‘ECMO alone’ patients subsequently received systemic or catheter directed thrombolysis after ECMO implantation as this would be reasonable practice in some centres and would significantly influence the interpretation of the results. 

The authors thank the reviewer for her/his comment. Patients who have only received an ECMO did not receive any kind of reperfusion therapy.

Other methodological issues 

 The multi variate logistic regression requires review in my opinion.

The authors thank the reviewer for her/his comment. With all due respect, the logistic regression model is not multivariate but univariate. (Please see above the previous explanations).

Of the five variables presented there is substantial overlap or association between them (pre-ECMO cardiac arrest and ECMO during CPR; thrombolysis failure is essentially synonymous with ECMO after thrombolysis in this study) and two of the variables appear to be the direct corollaries of each other (ECMO vs ECMO after thrombolysis AND ECMO after thrombolysis vs ECMO).

The authors thank the reviewer for her/his comment. We fully agree with the reviewer. ECMO only (vs ECMO + thrombolysis or catheter directed thromboaspiration) and ECMO + fibrinolytic or catheter-directed thromboaspiration (vs. ECMO only) concern the same analysis. Thus we delete line 4 in table 4. The new table 4 is as follow:

Table 4. Variables independently associated with 30-day survival status among ECMO-treated patients for massive PE.

Variables

OR

95%CI

p-value1

ECMO only (vs ECMO + thrombolysis or catheter directed thromboaspiration)

15.583

2.652 - 91.572

0.002

Thrombolysis failure

0.106

0.022-0.520

0.006

ECMO during cardiopulmonary resuscitation (vs. not)

0.174

0.039-0.773

0.022

Pre-ECMO cardia arrest (vs. not)

1Logistic regression model.

This does not appear to be robust. Additional factors that I would expect to be considered would be demographic variables such as age, markers of sickness severity (IS, lactate etc) and time to ECMO implantation. Even if these turn out not to be explanatory variables their inclusion makes the analysis and conclusion much more robust.

The authors thank the reviewer for her/his comment. Only the variables independently associated with 30-day survival status among ECMO-treated patients for massive PE are presented. We felt that presenting, habitual pertinent, variables that are, unfortunately, not associated with survival risks overloading the manuscript and misleading the reader.

The authors report mean values but then perform non-parametric tests. The sample size is very small and it seems unlikely that all of the variables reported are normally distributed. I recommend reporting median values.

The authors thank the reviewer for her/his comment. We agree with the reviewer. We changes that in the manuscript and tables in reporting median values with interquartile ranges, according to the small sample size.

The method only describes tests for unpaired data but then paired tests (Wilcoxon) are also reported – the method should be updated

The authors thank the reviewer for her/his comment. We agree with the reviewer. Changes have been made in the statistic paragraph accordingly:  “Results are presented as median and interquartile ranges (IQR) or numbers with percentages. Continuous variables were compared with the nonparametric Mann-Whitney U test or Wilcoxon signed ranks test as appropriate.”

Definitions should be made more clear please: e.g. i) what do the authors consider to be ‘absolute contraindications’ to thrombolysis – this should be explicit to allow readers to consider whether they agree as some contraindications may in fact be considered to be relative.

The authors thank the reviewer for her/his comment. Absolute Contraindications to fibrinolysis are listed in the 2019 ESC Guidelines for the diagnosis and management of acute pulmonary embolism developed in collaboration with the European Respiratory Society (ERS) [1] and include:

  • History of haemorrhagic stroke or stroke of unknown origin
  • Ischaemic stroke in previous 6 months
  • Central nervous system neoplasm
  • Major trauma, surgery, or head injury in previous 3 weeks
  • Bleeding diathesis
  • Active bleeding

These contraindications have been added in the paragraph “Competency in Medical Knowledge” as follow: “Massive pulmonary embolism (MAPE) is associated with increased risk of mortality related to profound obstructive cardiogenic shock and cardiac arrest. Thrombolysis, a treatment currently recommended in the absence of absolute contraindications, is not always effective and not without risk. Absolute contraindications to fibrinolysis are listed in the 2019 ESC Guideline for the diagnostic and management of acute pulmonary embolism and include History of haemorrhagic stroke or stroke of unknown origin, ischaemic stroke in previous 6 months, central nervous system neoplasm, major trauma, surgery, or head injury in previous 3 weeks, bleeding diathesis and active bleeding. VA-ECMO associated with full anticoagulation allows both rapid hemodynamics restoration and thromboembolic disease treatment.”

Ii) criteria for VA ECMO. 1. CS: are these AND, or OR criteria iii)

The authors thank the reviewer for her/his comment. It is actually OR as written in the paragraph highlighted in red: “The criteria for VA-ECMO implantation in the setting of MAPE were as follows: 1. refractory cardiogenic shock defined as evidence of tissue hypoxia (e.g., elevated blood lactate or extensive skin mottling) despite adequate volemia; severe RV failure; low cardiac index (≤ 2.1 L/min/m2) under inotropes; sustained hypotension despite high-dose vasoactive drugs infusion or 2. refractory cardiac arrest (no ROSC after 30 min of advanced resuscitation) with no-flow time below 5 min, low-flow time below 100 min, or effective external chest compressions with an end-tidal CO2 (EtCO2) > 1.5 kPa. ECMO exclusion criteria were age > 80 years old, malignancies with poor prognosis within 2 years or irreversible and disabling neurological pathologies and decisions to limit therapeutic interventions.

how did you define ‘RV failure’ or ‘high dose vasoactives’

The authors thank the reviewer for her/his comment. Severe RV failure has been defined in the text as follow: “severe RV failure defined as low cardiac output syndrome associated to echocardiographic signs of RV dilation, RV/LV dimension ration > 1.5 and septal flattering;”

The authors defined as well high dose vasoactive drugs in the text as follow: “sustained hypotension despite high-dose vasoactive drugs infusion (norepinephrine ≥ 0.3 µg /kg/min, epinephrine ≥ 0.3 µg/kg/min or dobutamine ≥ 10 µg/kg/min)”

  1. iv) haemorrhage – the finding of more haemorrhage in patients who receive thrombolysis  is important data but we need to know that this is robustly and objectively defined please.

The authors thank the reviewer for her/his comment. The definition of hemorrhage has been added in the text as follow: “Bleeding complications were reported using the Global Utilization of Streptokinase and TPA for Occluded arteries (GUSTO) classification [2]. Severe life-threatening hemorrhage was intracerebral bleeding or resulted in substantial hemodynamic compromise requiring treatment (GUSTO 1). GUSTO 2 defined moderate bleeding as the need for transfusion, whereas GUSTO 3 referred to other bleeding, not requiring transfusion or causing hemodynamic compromise.”

In the tables please provide the number of patients / data points that the values are based upon / including reporting of missing data. For example, the text refers to missing data from troponin but gives no further detail. In a retrospective study of emergency interventions I would expect there to be some missing data and how this is handled is important. 

The authors thank the reviewer for her/his comment. The sentence related to troponin and NT-ProBNP has been removed because there was too many missing data concerning these 2 parameters and the measurement’s methods have changed over the period of the study. All the other data presented in this study were present in the patient’s medical files, as these parameters are routinely measured for this kind of patients.

Minor comments:

The manuscript would benefit from a thorough proof read. Examples include:

Discrepancy between definition of groups in the introduction (Ln 63 – ECMO after thrombolysis) vs in the method (Ln 135-6 – ECMO after fibrinolysis or catheter aspiration)

The authors thank the reviewer for her/his comment. We corrected the sentence in the text: “We hereby present a retrospective cohort of patients with MAPE who were treated with VA-ECMO alone as a part of advanced life support or as rescue treatment after a failed attempt at systemic thrombolysis and/or catheter thromboaspiration.”

Moreover, the present manuscript has been proof edited by Am J Expert (please look to the certificate)

Ln 56 – formatting of references

The authors thank the reviewer for her/his comment. We formatted this reference as “18”: 18. Abraham, P.; Arroyo, D.A.; Giraud, R.; Bounameaux, H.; Bendjelid, K. Understanding haemorrhagic risk following thrombolytic therapy in patients with intermediate-risk and high-risk pulmonary embolism: a hypothesis paper. Open Heart 2018, 5, e000735, doi:10.1136/openhrt-2017-000735.

Ln n147 – instauration = institution

The authors thank the reviewer for her/his comment. We changed the word in the sentence as follow:  “Accordingly, before ECMO institution, both mean systolic arterial (71 ± 43 mmHg) and mean arterial (51 ± 30 mmHg) blood pressures and pre-ECMO heart rate (80 ± 50 beats/min) were low.”

Tables: please include abbreviations in full in a legend

The authors thank the reviewer for her/his comment. We added full legend in the tables.

There is a discrepancy between the heart rate value presented in the text (Ln 148) and in Table 1

The authors thank the reviewer for her/his comment. We changes the values in the tables and presented the data as median and interquartile ranges.

TAPSE appears to be presented as cm when the values suggest should be mm

The authors thank the reviewer for her/his comment. We changes TAPSE in mm.

There is a lot of duplication of results in the text that already exist in tables and this could be revised

The authors thank the reviewer for her/his comment. We removed some redundancy parameters in the “results” paragraph as follow: “During the 10-year period of the study, 36 patients (27 males (75%); median age 57 (IQR 23) presenting with MAPE received VA-ECMO. Table 1 displays the description of the complete cohort. Thirty (83.3%) patients had predisposing factors for venous thromboembolism. Twenty-two (61.7%) patients presented with cardiac arrest—all with a no-flow time of zero min and a low-flow time of 28 ± 30 min. Thirteen (36.1%) patients received ECMO as part of advanced cardiopulmonary resuscitation (eCPR). Accordingly, before ECMO institution, both median systolic arterial (74 mmHg (IQR 54)) and mean arterial (59 mmHg (IQR 35)) blood pressures were low. Mean values for the acid-base status reflected this situation with a low pH (7.08 (IQR 0.38)), high blood lactate (8.3 mmol/L (IQR 11.1)) and low bicarbonate (15.2 mmol/L (IQR 8)).”

A few abbreviations are introduced without prior explanation (e.g ‘CO’ Ln 236, ‘ESC’ Ln 240)

The authors thank the reviewer for her/his comment. These abbreviations have been corrected in the text: “A low level of cardiac output causes desaturation of mixed venous blood. Areas of reduced flow in the obstructed pulmonary arteries, associated with areas of overflow in the capillary bed served by unobstructed pulmonary vessels, cause a ventilation/perfusion mismatch, which contributes to hypoxaemia [3] and an increase in dead space. ECMO is now recommended by the latest Guidelines of the European Society of Cardiology (ESC) as temporary mechanical circulatory support in patients with MAPE and circulatory collapse or cardiac arrest [1], offering full cardiopulmonary support [4], as shown by our results. Indeed, pH, blood lactate level and IS significantly improved 24 h after ECMO implantation.”

The discussion would benefit from some revision to make it more focused and to avoid some apparent confusions. E.g. i) The discussion about surgical embolectomy does not follow from this study as there is no data at all about this treatment (Ln291-302) ii) Ln 283-4 –

the conclusion about mortality being associated with failed thrombolysis does not naturally follow from the prior discussion iii) Ln. 320 – the factors listed are not causes of death but are factors that associate with death

The authors thank the reviewer for her/his comment. According to these remarks, we change the order of the paragraphs in the discussion for a better continuity. However, we keep the small paragraph about surgical embolectomy as some teams use this therapy in association with VA-ECMO. Our opinion differs from this practice. Indeed, we believe that ECMO, including hemodynamic stabilization of the most unstable patients, associated with treatment with a full heparinization, allows the resolution of MAPE, without having to perform major surgery, which in addition, allows only partial removal of larger clots located proximally.

Respectfully, I think the discussion of weaknesses ought be expanded given the material issues identified above, unless they can be resolved to show more robustly that the conclusion of the study is robust.

The authors thank the reviewer for her/his comment. A paragraph about PERT has been added in the discussion as follow: “Given patient heterogeneity, multitude of available treatment modalities, and lack of consensus guidelines, treatment strategies for these complex patients with MAPE are often debated. Multidisciplinary Pulmonary Embolism Response Teams (PERT) are dedicated to address this lack of consensus [5]. An analysis among 769 consecutive inpatients with PE was conducted to compare outcomes of all patients with PE before and after PERT availability. Results shows that PERT era patients had lower rates of major or clinically relevant non-major bleeding (17.0% vs 8.3%, p = 0.002), shorter time-to-therapeutic anticoagulation (16.3 hour vs 12.6 hour, p = 0.009). There was also a significant decrease in 30-day/inpatient mortality (8.5% vs 4.7%, p = 0.03), particularly in patients with MAPE (mortality 10.0% vs 5.3%, p = 0.02) [6]. As in our center, the availability of multidisciplinary PERT certainly allows improved outcomes including 30-day mortality.”

Moreover, the report by Al-Bawardy et al (DOI: 10.1177/0267659118786830), not included in the reference, list report 31% 30-day mortality in a short series of 13 ECMO treated cases, most of them being treated also by thrombolysis. This study has been cited and discussed as follow: “In a consecutive cohort of thirteen patients with confirmed MAPE for whom PERT was activated and selected patients treated with ECMO confirmed that patients with MAPE who suffer a cardiac arrest have high morbidity and mortality [7].”

REFERENCE:

  1. Konstantinides, S.V.; Meyer, G.; Becattini, C.; Bueno, H.; Geersing, G.J.; Harjola, V.P.; Huisman, M.V.; Humbert, M.; Jennings, C.S.; Jimenez, D., et al. 2019 ESC Guidelines for the diagnosis and management of acute pulmonary embolism developed in collaboration with the European Respiratory Society (ERS). Eur Heart J 2020, 41, 543-603, doi:10.1093/eurheartj/ehz405.
  2. Investigators, G.A. The effects of tissue plasminogen activator, streptokinase, or both on coronary-artery patency, ventricular function, and survival after acute myocardial infarction. The New England journal of medicine 1993, 329, 1615-1622, doi:10.1056/NEJM199311253292204.
  3. Burrowes, K.S.; Clark, A.R.; Tawhai, M.H. Blood flow redistribution and ventilation-perfusion mismatch during embolic pulmonary arterial occlusion. Pulm Circ 2011, 1, 365-376, doi:10.4103/2045-8932.87302.
  4. Tahir, U.A.; Carroll, B.; Pinto, D.S. Massive pulmonary embolism: embolectomy or extracorporeal membrane oxygenation? Curr Opin Crit Care 2019, 25, 630-637, doi:10.1097/MCC.0000000000000660.
  5. Kabrhel, C.; Jaff, M.R.; Channick, R.N.; Baker, J.N.; Rosenfield, K. A multidisciplinary pulmonary embolism response team. Chest 2013, 144, 1738-1739, doi:10.1378/chest.13-1562.
  6. Chaudhury, P.; Gadre, S.K.; Schneider, E.; Renapurkar, R.D.; Gomes, M.; Haddadin, I.; Heresi, G.A.; Tong, M.Z.; Bartholomew, J.R. Impact of Multidisciplinary Pulmonary Embolism Response Team Availability on Management and Outcomes. Am J Cardiol 2019, 124, 1465-1469, doi:10.1016/j.amjcard.2019.07.043.
  7. Al-Bawardy, R.; Rosenfield, K.; Borges, J.; Young, M.N.; Albaghdadi, M.; Rosovsky, R.; Kabrhel, C. Extracorporeal membrane oxygenation in acute massive pulmonary embolism: a case series and review of the literature. Perfusion 2019, 34, 22-28, doi:10.1177/0267659118786830.

Reviewer 2 Report

This is an interesting case series showing excellent results and suggesting therapeutic attitudes that could improve the survival of such a complex group of patients. I just have a few minor comments and I thank the authors for reading and considering them.

  1. Standardising the care of MAPE patients is paramount. The 2019 European Society of Cardiology guidelines on management of intermediate and high-risk PE include implementing institutional PE response teams and one of the recommendations. I haven’t seen in the text if the authors’ institutions have implemented PERTs. A comment on that at the Methods or Discussion sections would be welcome.
  2. The report by Al-Bawardy et al (DOI: 10.1177/0267659118786830), not included in the reference, list report 31% 30-day mortality in a short series of 13 ECMO treated cases, most of them being treated also by thrombolysis. Although this series is not comparable to yours, due to controversial results you could include this reference in your manuscript and discuss their results.
  3. Including 95%CIs in figure 2 could help the reader to evaluate the consistency of results due to small numbers in both series of cases.

Author Response

Dear Professor Bendjelid

Revised version R1- jcm-1314069: Can VA-ECMO Be Used as an Adequate Treatment in Massive Pulmonary Embolism?

We thank you for the careful review of our manuscript and for the constructive criticisms. The suggestions were very helpful and we appreciated them. We have revised the manuscript as suggested.

You will find in attached files our point-by-point responses to the comments of the reviewers. We also include a revised version of our manuscript, containing a document with changes highlighted in red.

All authors have contributed significantly to the present work. They have read and approved submission of the manuscript,

We thank you by advance for reconsidering our manuscript and hope that you will now find it suitable for publication in Journal of Clinical Medicine. Additionally, we thank you for your generous consideration.

Looking forward to hearing from you.

Raphael Giraud, MD, PhD

Reviewer 2

This is an interesting case series showing excellent results and suggesting therapeutic attitudes that could improve the survival of such a complex group of patients. I just have a few minor comments and I thank the authors for reading and considering them.

  1. Standardising the care of MAPE patients is paramount. The 2019 European Society of Cardiology guidelines on management of intermediate and high-risk PE include implementing institutional PE response teams and one of the recommendations. I haven’t seen in the text if the authors’ institutions have implemented PERTs. A comment on that at the Methods or Discussion sections would be welcome.

The authors thank the reviewer for her/his comment. A sentence about PERT has been added in the “method” paragraph. “The criteria for VA-ECMO implantation in the setting of MAPE were decided by a pulmonary embolism response teams (PERTs) including intensivists, cardiac surgeons, cardiologist, anaesthesiologists and angiologists and are the following:”

Moreover, a paragraph about PERT has been added in the discussion as follow: “Given patient heterogeneity, multitude of available treatment modalities, and lack of consensus guidelines, treatment strategies for these complex patients with MAPE are often debated. Multidisciplinary Pulmonary Embolism Response Teams (PERT) are dedicated to address this lack of consensus [1]. An analysis among 769 consecutive inpatients with PE was conducted to compare outcomes of all patients with PE before and after PERT availability. Results shows that PERT era patients had lower rates of major or clinically relevant non-major bleeding (17.0% vs 8.3%, p = 0.002), shorter time-to-therapeutic anticoagulation (16.3 hour vs 12.6 hour, p = 0.009). There was also a significant decrease in 30-day/inpatient mortality (8.5% vs 4.7%, p = 0.03), particularly in patients with MAPE (mortality 10.0% vs 5.3%, p = 0.02) [2]. As in our center, the availability of multidisciplinary PERT certainly allows improved outcomes including 30-day mortality.”

  1. The report by Al-Bawardy et al (DOI: 10.1177/0267659118786830), not included in the reference, list report 31% 30-day mortality in a short series of 13 ECMO treated cases, most of them being treated also by thrombolysis. Although this series is not comparable to yours, due to controversial results you could include this reference in your manuscript and discuss their results.

The authors thank the reviewer for her/his comment. This study is now cited and discussed as follow: “In a consecutive cohort of thirteen patients with confirmed MAPE for whom PERT was activated and selected patients treated with ECMO confirmed that patients with MAPE who suffer a cardiac arrest have high morbidity and mortality [3].”

  1. Including 95%CIs in figure 2 could help the reader to evaluate the consistency of results due to small numbers in both series of cases.

The authors thank the reviewer for her/his comment. 95%CIs in figure 2 have been added.

REFERENCE :

  1. Kabrhel, C.; Jaff, M.R.; Channick, R.N.; Baker, J.N.; Rosenfield, K. A multidisciplinary pulmonary embolism response team. Chest 2013, 144, 1738-1739, doi:10.1378/chest.13-1562.
  2. Chaudhury, P.; Gadre, S.K.; Schneider, E.; Renapurkar, R.D.; Gomes, M.; Haddadin, I.; Heresi, G.A.; Tong, M.Z.; Bartholomew, J.R. Impact of Multidisciplinary Pulmonary Embolism Response Team Availability on Management and Outcomes. Am J Cardiol 2019, 124, 1465-1469, doi:10.1016/j.amjcard.2019.07.043.
  3. Al-Bawardy, R.; Rosenfield, K.; Borges, J.; Young, M.N.; Albaghdadi, M.; Rosovsky, R.; Kabrhel, C. Extracorporeal membrane oxygenation in acute massive pulmonary embolism: a case series and review of the literature. Perfusion 2019, 34, 22-28, doi:10.1177/0267659118786830.
